# Effect of d-cycloserine on fear extinction training in adults with social anxiety disorder

**Stefan G. Hofmann** [1] *, **Santiago Papini**[2], **Joseph K. Carpenter**[1], **Michael W. Otto**[1], **David Rosenfield**[3], **Christina D. Dutcher**[2], **Sheila Dowd**[4], **Mara Lewis**[1], **Sara Witcraft** [5], **Mark H. Pollack**[4], **Jasper A. J. Smits**[2]

**1** Department of Psychological and Brain Sciences, Boston University, Boston, Massachusetts, United States of America, **2** Department of Psychology, University of Texas at Austin, Austin, Texas, United States of America, **3** Department of Psychology, Southern Methodist University, Dallas, Texas, United States of America, **4** Department of Psychiatry, Rush University Medical Center, Chicago, Illinois, United States of America, **5** Department of Psychology, University of Mississippi, Oxford, Mississippi, United States of America

* shofmann@bu.edu

**Data Availability Statement:** All relevant data are within the paper and its Supporting Information files.

## Abstract

Preclinical and clinical data have shown that D-cycloserine (DCS), a partial agonist at the N-methyl-d-aspartate receptor complex, augments the retention of fear extinction in animals and the therapeutic learning from exposure therapy in humans. However, studies with non-clinical human samples in de novo fear conditioning paradigms have demonstrated minimal to no benefit of DCS. The aim of this study was to evaluate the effects of DCS on the retention of extinction learning following de novo fear conditioning in a clinical sample. Eighty-one patients with social anxiety disorder were recruited and underwent a previously validated de novo fear conditioning and extinction paradigm over the course of three days. Of those, only 43 (53%) provided analyzable data. During conditioning on Day 1, participants viewed images of differently colored lamps, two of which were followed by with electric shock (CS+) and a third which was not (CS-). On Day 2, participants were randomly assigned to receive either 50 mg DCS or placebo, administered in a double-blind manner 1 hour prior to extinction training with a single CS+ in a distinct context. Day 3 consisted of tests of extinction recall and renewal. The primary outcome was skin conductance response to conditioned stimuli, and shock expectancy ratings were examined as a secondary outcome. Results showed greater skin conductance and expectancy ratings in response to the CS+ compared to CS- at the end of conditioning. As expected, this difference was no longer present at the end of extinction training, but returned at early recall and renewal phases on Day 3, showing evidence of return of fear. In contrast to hypotheses, DCS had no moderating influence on skin conductance response or expectancy of shock during recall or renewal phases. We did not find evidence of an effect of DCS on the retention of extinction learning in humans in this fear conditioning and extinction paradigm.

**Funding:** This study was funded as a multi-site linked R34 grant (R34MH099311, R34MH099318, R34MH099309; ClinicalTrials.gov identifier: NCT02066792) by the National Institute of Mental Health (Principal Investigators: SGH, MHP, and JAJS).

**Competing interests:** The authors have read the journal's policy and the authors have the following competing interests: SGH receives financial support from the Alexander von Humboldt Foundation and compensation for his work as editor from SpringerNature and the Association for Psychological Science. He also receives compensation for this role as an advisor from the Palo Alto Health Sciences and for his work as a Subject Matter Expert from John Wiley & Sons, Inc. and SilverCloud Health, Inc. MWO receives financial support for his role as chair on the scientific advisory board for Big Health. Ltd. MP receives financial support for Consultation and his role on the advisory boards for the following Almatica Pharma, Aptinyx, Brackett Global, Brainsway, EMA Wellness, Seelos Therapeutics, Sophren Therapeutics; Research Grants: NIH, Janssen; Equity: Argus, Doyen Medical, Medavante, Mensante Corporation, Mindsite, Targia Pharmaceuticals; Royalty/patent: SIGH-A, SAFER interviews. JAJS receives compensation for his role as a consultant to Big Health, Ltd. This does not alter the authors' adherence to PLOS ONE policies on sharing data and materials.

## Introduction

Exposure-based treatment for the anxiety-related disorders offers some of the strongest treatment outcomes in the literature [1,2]. Nonetheless, a proportion of patients fail to respond adequately to these treatments and others face relapse after treatment [3,4]. Accordingly, a number of efforts are underway to strengthen the efficacy and durability of extinction learning from exposure therapy [5].

One of these strategies is to use D-cycloserine (DCS)—a partial agonist at the N-methyl-d-aspartate receptor—as a way to augment the retention of therapeutic learning from exposure [6]. Although results across individual clinical trials have been variable (e.g., [7,8]), a recent meta-analysis indicates that across disparate clinical trials of anxiety disorders, DCS augmentation of exposure therapy offers advantages on the order of a small effect size ($d = 0.25$) for enhancing early response to treatment relative to placebo [9]. Nonetheless, there is evidence of significant moderation of these studies by the diagnostic target of interventions, with evidence of larger augmentation effects in social anxiety compared to other anxiety disorders [9]. In addition to diagnostic variability, DCS augmentation studies differ in the amount of exposure therapy offered, the elements of treatment in addition to exposure, and the amount and timing of DCS administrations relative to the start of exposure treatment [6]. Any of these factors could introduce variability into estimates of the efficacy of DCS augmentation.

In order to understand the reasons for this variability, DCS augmentation of fear extinction has been studied in human de novo fear conditioning paradigms. Potential advantages of this approach are that it offers: (1) a close analogue to the animal conditioning studies that have shown DCS efficacy [10–15] and which served as the basis of the trials that tested this clinical strategy, (2) experimental control of the degree of acquisition and extinction training provided, (3) lower study costs, and (4) consistency with a Research Domain Criteria (RDoC) research framework for encouraging precise and replicable measurement of basic processes as a strategy for advancing mechanistic understanding of interventions [16], in this case fear extinction as it relates to exposure therapy efficacy. Despite this hope, and despite evidence of successful DCS augmentation in clinical trials, initial human studies using de novo fear conditioning paradigms have shown minimal to no effect of DCS on the retention of extinction learning. Specifically, studies examining the effects of DCS augmentation on extinction recall and fear renewal have consistently had null findings [17–20], though some evidence of reduced reinstatement has been found [19,20].

The purpose of this study was to evaluate the efficacy of DCS for enhancing the effects of extinction training on extinction retention as evaluated in a human de novo fear conditioning paradigm [21–22]. This study incorporated a number of innovations to address limitations in the DCS literature to date. First, rather than studying healthy control participants, we employed a population for whom DCS clinical effects have been particularly strong [7]: outpatients with social anxiety disorder. Second, we reduced cues for higher-order processing (i.e. shock expectancy ratings administered during CS presentations), while retaining some assessment of explicit knowledge of the fear contingency in the form of retrospective expectancy ratings administered at the end of each experimental phase. This enabled us to evaluate post-hoc whether DCS effects on skin conductance response (SCR) were mirrored by expectancy ratings. Third, to provide a direct test of extinction effects that have an analogue to clinical fears, we assessed DCS vs. placebo augmentation effects only in individuals who demonstrated adequate acquisition of de novo fears (indeed, among both anxious and healthy samples a substantial proportion of participants may fail to show fear acquisition on skin conductance measures [18,23,24], and there is evidence that poor skin conductance conditioning reflects hypoactivation of brain regions involved in fear learning and expression [25]). Our primary hypothesis

was that DCS would augment de novo fear extinction learning of SCR through increased retention of extinction during a recall and renewal phase occurring 24 hours later. Prior to study initiation, hypotheses (i.e., DCS enhancement of extinction recall and reduction of fear renewal) were published in Hofmann et al. [26]. Prior to data analysis we made several modifications to the analytic approach described in Hofmann et al. (2015) [23] to be consistent with the latest methodological advancements and recommendations. Specifically, we used continuous decomposition analysis to extract skin conductance responses and we tested the pre-specified hypotheses in ANOVA that included a term for contrasts between stimuli, as opposed to subtracting CS- SCRs from CS+ SCRs prior to analyses [27]. Another modification was to omit the prespecified "Extinction Retention Index" (ERI) analysis in light of a recent publication [28] which outlined theoretical and procedural problems with its operationalization, including the existence of 16 different calculations of the ERI in the literature.

## Methods

The study was approved as Human Subject Research by the respective Institutional Review Boards of Boston University, Rush University Medical Center, Southern Methodist University, and University of Texas at Austin. All participants provided written informed consent to participate.

### Participants

Participants ($N = 81$) consisted of a subset of patients enrolled in a multisite clinical trial for social anxiety disorder (SAD) [26] taking place at Boston University, University of Texas at Austin, and Rush University Medical Center. Prior to beginning the clinical trial, participants elected to undergo the present laboratory extinction study, and were compensated $120 for their time. Inclusion criteria consisted of (1) a primary diagnosis of SAD as defined by DSM-5 criteria, (2) a total score $\geq 60$ on the clinician-administered Liebowitz Social Anxiety Scale (LSAS) [29], (3) passing of a medical examination without any detection of conditions that would contraindicate the administration of DCS, including pregnancy, lactation or a history of seizures; (4) at least 18 years of age. Exclusion criteria included (1) a lifetime history of bipolar, a psychotic disorder, organic brain syndrome, mental retardation, or other potentially interfering cognitive dysfunction; (2) eating disorder, posttraumatic stress disorder, obsessive-compulsive disorder, substance abuse or dependence (other than nicotine), or significant suicidal ideation or behaviors in the past 6 months; (3) concurrent psychotropic medication within the past 2 weeks; (4) current psychotherapy initiated in the prior three months, or ongoing psychotherapy directed toward treatment of SAD; (5) prior non-response to exposure therapy; and (6) a history of head trauma causing loss of consciousness, seizure or ongoing cognitive impairment.

### Procedure

Participants underwent a diagnostic clinical interview, administration of the LSAS, and a medical examination to determine eligibility for the clinical trial portion of the study. A semi-structured assessment of depression, the Montgomery Asberg Depression Rating Scale (MADRS) [30], was also administered at this time. If eligible, participants were invited to complete the laboratory portion of the study, which took place over three consecutive days. Eighty-one of 172 patients in the clinical trial elected to participate in the laboratory experiment. Experimental procedures were based on a previously validated and widely used fear conditioning and extinction paradigm [21]. On Day 1, participants went through habituation and conditioning procedures. On Day 2, participants were randomly assigned to receive either 50 mg DCS or

placebo (PBO), which was administered in a double-blind manner 1 hour prior to extinction training. Day 3 consisted of a test of extinction recall and renewal.

## Stimuli and experimental protocol

Participants viewed images on a computer monitor while two 9-mm (sensor diameter) Sensor Medics Ag/AgCl recording electrodes were attached to their left hand to measure skin conductance response (SCR) as the primary dependent variable. Two stimulating electrodes were also attached to the second and third fingers on the right hand to deliver a 500-millisecond electric shock, which served as the unconditioned stimulus (US). Shock was generated by a Coulbourn Transcutaneous Aversive Finger Stimulator, and mean shock intensity was 1.86 Milliamperes (SD = 1.56). At the University of Texas-Austin site, a BIOPAC MP150 Psychophysiological Recording Apparatus (BIOPAC Systems, Inc., USA), was used, and data were acquired using AcqKnowledge 4.0 software. At Boston University and Rush University, psychophysiological data were recorded with custom equipment made by James Long Company, Caroga Lake, NY, and the data-acquisition program Snap-Master for Windows. Across sites, the sampling rate was 1000 Hz. Prior to the presentation of any images, participants were exposed to increasing intensities of shock until they judged it to be "highly annoying but not painful," and this intensity was used for conditioning. The electrodes were attached to the fingers on all three days, even though the US was only delivered on Day 1. Images consisted of photographs of two distinct rooms, one with a desk and computer ("threat context" used in the conditioning phase where shock was delivered) and the other with a bookshelf ("safe context" used in extinction and recall phases where no shocks were delivered), both of which contained the same unlit lamp (Fig 1). During each trial, the context was presented for 3 seconds, and then the lamp "switched on" and became either a blue, red or yellow light for 6 seconds. These colored lamps formed the conditional stimuli (CSs), with two of the colors being followed immediately by shock (CS+) during Conditioning on Day 1, while the third was not (CS-). One CS+ was randomly assigned to be the CS+E (extinguished CS+) and was used for Extinction on Day 2, while the other (unextinguished CS+, or CS+U) was not seen again until the Recall and Renewal phases on Day 3. Thus, on Day 3, responding was evaluated both to the extinction stimulus (CS+E) as well as a perceptually similar but unextinguished CS+ (CS+U) that served as a test of extinction generalization. The inter-trial interval consisted of a black screen that lasted between 12 and 18 seconds. Throughout all three days, stimulus order within each block was pseudo-randomized such that the same stimulus never appeared more than three times in a row, and each block always began with a CS+ (reinforced during conditioning). Stimulus order and the color of the CS+ and the CS- were counterbalanced across participants.

A schematic of the experimental paradigm can be seen in Fig 1. During Habituation on Day 1, the three CSs were presented in each of the two contexts (six trials total) to familiarize participants with the stimuli. Participants then immediately went through Conditioning, which consisted of two blocks of 16 trials, each with eight presentations of the CS- interspersed with eight presentations of either the CS+E or the CS+U. Five of the eight CS+ presentations were followed immediately by the US (62.5% reinforcement). This reinforcement rate was used to replicate procedures from the previously validated paradigm used for this study [21,22], and to prevent the rapid extinction seen in protocols with 100% reinforcement [27,31]. All stimuli presented during Conditioning were presented in the threat context (e.g., desk and computer).

During Extinction on Day 2, all stimuli were presented in the safe context. The CS- and CS+E were each presented 16 times, but in contrast to Day 1 the CS+E was never followed by shock (US). The first stimulus presented during Extinction was always the CS+.

| A) Habituation (Day 1) | B) Conditioning (Day 1) | C) Extinction (Day 2) | D) Recall (Day 3) | E) Renewal (Day 3) |
|---|---|---|---|---|
| 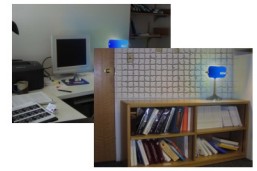 | 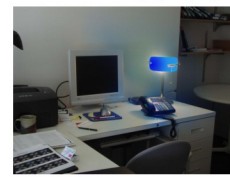 | 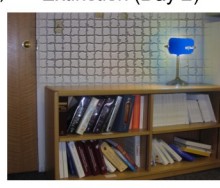 | 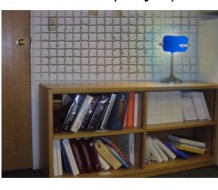 | 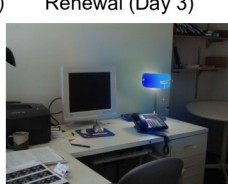 |
| presented once in each context | threat context | safe context | safe context | threat context |
| 2 CS- \| 2 CS+E \| 2 CS+U | 16 CS- \| 8 CS+E \| 8 CS+U<br>10 shocks (5 for each CS+) | 16 CS- \| 16 CS+E<br>*50 mg of DCS or PBO*<br>*1h before extinction* | 16 CS- \| 8 CS+E \| 8 CS+U | 16 CS- \| 8 CS+E \| 8 CS+U |

**Fig 1. Schematic of experimental paradigm.** In the habituation phase on Day 1, stimuli (i.e., lamp "turning on" to show one of three colors) were presented across the two contexts (A). In the conditioning phase on Day 1 (B), the color of the lamp light served as a conditioned Stimulus. Three lights were presented randomized and counterbalanced across participants. Two lights were paired with shock at a 62.5% reinforcement rate and served as the conditioned stimuli that either underwent extinction on Day 2 (CS+E) or remained unextinguished until Day 3 (CS+U). A third color, never paired with shock, provided a reference for differential conditioning (CS-). In the extinction phase on Day 2 (C), the CS+E and CS- were presented without shock in a different (safe) context. One hour prior to extinction, participants took either DCS or PBO. In the recall (D) and renewal (E) phases on Day 3, all three stimuli were presented without shock to test between-session extinction retention and generalization of extinction to the threat context, respectively.

On Day 3, the CS-, CS+E and CS+U were each presented first in the safe context as a test of extinction recall. Similar to conditioning on Day 1, this involved two blocks of 16 trials, each with 8 presentations of the CS- and 8 presentations of either the CS+E or CS+U. Immediately following, participants underwent a test of renewal, which involved the same procedures as recall except that CS's were presented within threat context. For both recall and renewal, the order of CS+E and CS+U blocks were counterbalanced across participants.

At the end of each phase, participants completed questions about which colored lamps they saw and which colors were followed by a shock in order to assess contingency awareness. They also answered questions regarding US expectancy, specifically: "on a scale from 1 (not at all) to 5 (very much), how much were you expecting to be shocked for the [first or last] presentation of the [red, blue or yellow] lamp?", with separate questions for the first and last presentation of each color of lamp seen during the phase. Retrospective US expectancy was investigated as a secondary outcome.

## SCR preprocessing

Skin conductance response (SCR) data were preprocessed and extracted in Ledalab software version 3.4.9 using the following approach: (1) raw skin conductance data were inspected and (where possible) corrected for gross motion artifacts and poor signal quality; (2) SCL data were downsampled to 10 Hz and smoothed using an adaptive Gaussian approach; (3) SCRs within the 6 s stimulus window were extracted using continuous decomposition analysis [32]; (4) square-root transformation was applied to normalize SCRs; (5) within participants, extreme outliers, defined as individual SCRs that were 3 SDs greater than the participant's mean SCR amplitude, were removed; (6) for the conditioning phase, the last four SCRs of each stimulus were averaged to calculate late conditioning, and for the remaining phases, the first two (early) and last two (late) trials were averaged; (7) participants that did not demonstrate good differential SCR conditioning, defined as SCR to the CS+E that was greater than the CS- by at least $0.1 \sqrt{\mu S}$ in the late conditioning stage, were removed from the analysis. This approach is consistent with our previous work [23,24], and was done to ensure that participants included in the analysis demonstrated adequate fear learning that could meaningfully be subjected to extinction and renewal procedures in the subsequent phases of the study (see also Marin et al., [25]). The cutoff of $0.1 \sqrt{\mu S}$ is commonly used in literature on *de novo* fear conditioning [23,24,33–36]. Since hypotheses were tested within each phase, participants were not

excluded from analysis in one phase when they had incomplete data in another phase, which resulted in minor variations in sample size across phases.

## Statistical analyses

For the conditioning phase, mean SCRs and US expectancy ratings during the late stage were compared in a Group (DCS, PBO) × Stimulus (CS+E, CS+U, CS-) ANOVA. For the extinction phase, SCR and US expectancy ratings were compared in a Group (DCS, PBO) × Stimulus (CS +E, CS-) × Stage (early, late) ANOVA. For the recall and renewal phases, SCR and US expectancy ratings were compared in a Group (DCS, PBO) × Stimulus (CS+E, CS+U, CS-) × Stage (early, late) ANOVA. When statistical assumptions were violated, corresponding corrections were applied and reported. Test statistics, uncorrected $p$-values, and partial eta-squared ($\eta^2_p$) effect sizes are provided for all main effects and interactions. Where applicable, significant results were followed up with *post hoc* pairwise comparisons and statistics with Cohen's $d$ effect sizes are provided.

## Results

Of the 81 participants who participated in the fear conditioning paradigm, 12 had unusable SCR data due to equipment malfunctioning. Of the 69 participants with usable SCR data, 43 demonstrated good differential conditioning between the CS+E and the CS- during late conditioning, a similar proportion to what has been reported previously for anxious samples [23,24]. Total LSAS score, $t(60.64) = 1.31$, $p = 0.198$, MADRS score, $t(58.57) = -0.13$, $p = 0.896$, and US intensity (i.e. individually selected shock level), $t(64) = -.35$, $p = .725$ were not significantly different between participants that did and did not show differential SCR conditioning, nor was differential US expectancy (CS+E minus CS-), $t(66) = -0.59$, $p = 0.559$, conditioners: $M = 2.33$, $SD = 1.34$; non-conditioners: $M = 2.11$, $SD = 1.70$, or likelihood of contingency awareness, $\chi^2 (1) = 2.42$, $p = 0.120$, at the end of the conditioning phase. Following the recommendations of Lonsdorf et al. [27], we performed sensitivity analyses to determine whether exclusion of non-conditioners influenced results. No differential effects were obtained relative to those reported below, and we report these results as supplementary material (see S2 File).

Participants did not significantly differ across experimental groups in severity of social anxiety as based on the LSAS (DCS: $M = 84.90$, $SD = 19.74$; PBO: $M = 80.45$, $SD = 16.42$; $t(41) = 0.81$, $p = 0.43$) or depression as measured by the MADRS (DCS: $M = 8.55$, $SD = 7.48$; PBO: $M = 14.52$, $SD = 12.23$; $t(41) = 1.95$, $p = 0.06$). Participants in the DCS group ($n = 22$) had a mean age of 25.24 ($SD = 4.82$), 59.1% were female, 31.8% identified as Hispanic or Latino, and had a racial breakdown as follows: 50.00% White, 4.55% Black or African-American, 22.73% Asian, and 22.73% other. In the PBOgroup ($n = 21$), mean age was 29.71 ($SD = 11.34$), 52.4% of participants were female, 23.8% identified as Hispanic or Latino, and racial breakdown was as follows: 57.14% White, 19.05% Black or African-American, and 23.81% other. There were no significant differences across groups in demographic variables (all $p$s > 0.10).

## SCR results

Fig 2 shows mean SCRs during each phase of the experiment and Table 1 shows complete statistics of the corresponding ANOVAs. Significant results are summarized below.

**Conditioning.** In the conditioning phase, there was a significant main effect of Stimulus, $F(2, 82) = 36.86$, $p < .001$, $\eta^2_p = .47$, reflecting a large effect size. Post-hoc tests indicated that mean SCR was greater for the CS+E than the CS-, $t(82) = 8.10$, $p < .001$, $d = 2.471$, and greater for the CS+U than the CS-, $t(82) = 6.51$, $p < .001$, $d = 1.985$, but not significantly different between the CS+E and CS+U, $t(82) = 1.60$, $p = .11$, $d = 0.486$. As expected at this pre-treatment

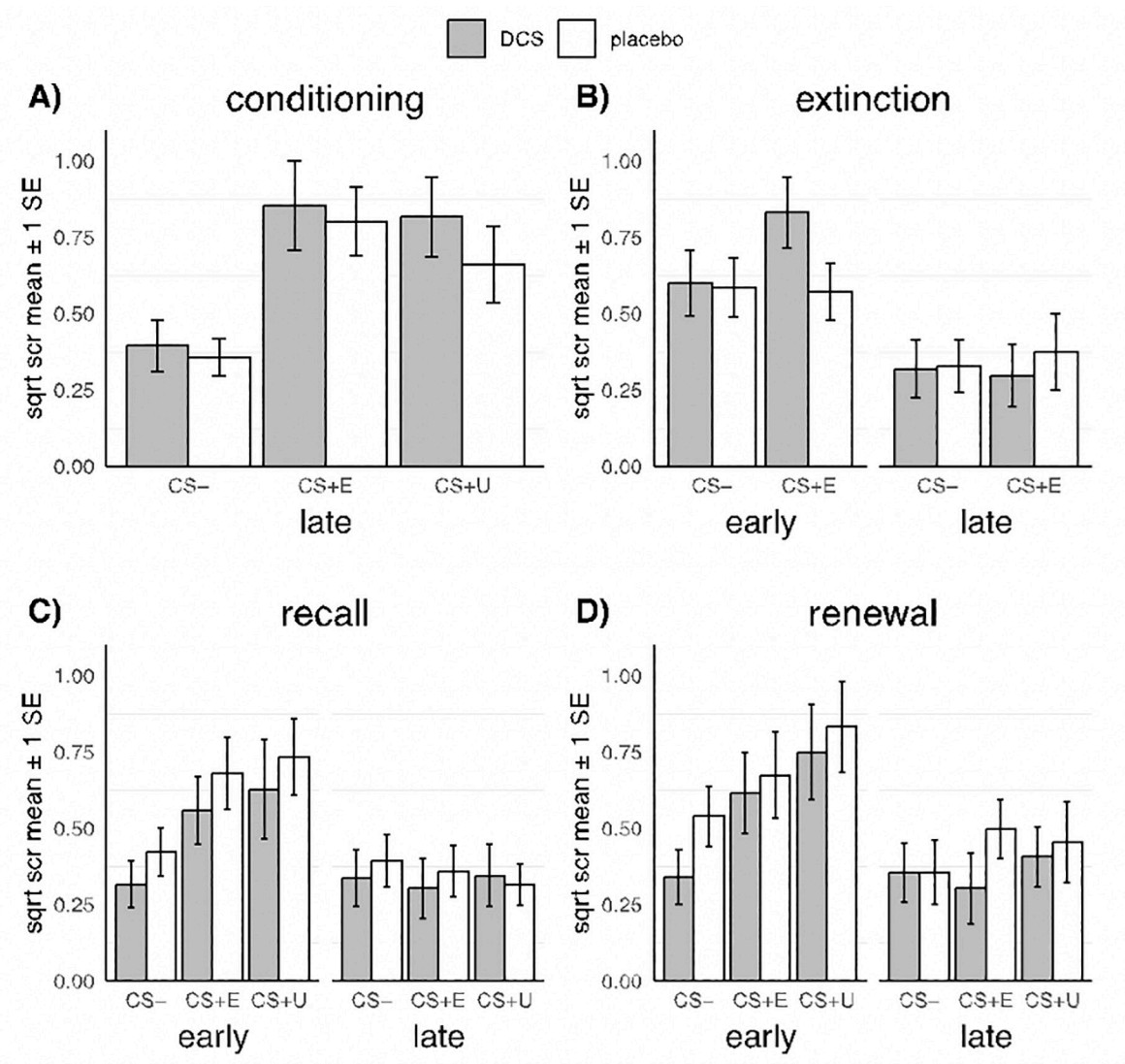

**Fig 2. Conditioned skin conductance responses (SCR) across all phases of the experiment.** Note. DCS was administered 1 hour prior to extinction. PBO = placebo group, DCS = d-cycloserine, SCR = skin conductance response, SE = 1 standard error, CS- = stimulus that was not paired with shock, CS+E = stimulus that was paired with shock during conditioning and presented in the extinction phase, CS+U = stimulus that was paired with shock during conditioning but not presented in the extinction phase.

stage there was no evidence that DCS and PBO groups differed in their overall or CS-specific SCRs.

**Extinction.** In the extinction phase, a significant main effect of Stage provided evidence of an overall decrease in mean SCR from the early to the late stage of extinction, $F(1, 36) = 21.52$, $p < .001$, $\eta^2_p = .37$. All other effects were nonsignificant; there was no evidence of Group effects, consistent with the hypothesis that the effect of DCS should take place on the consolidation of extinction learning rather than the amount of in-session learning.

**Recall.** In the recall phase, there was a significant effect of Stimulus, $F(1.39, 51.44) = 4.72$, $p = .023$, $\eta^2_p = .11$, Stage, $F(1, 37) = 31.86$, $p < .001$, $\eta^2_p = .46$, and a Stimulus × Stage interaction $F(1.62, 60.10) = 5.93$, $p = .007$, $\eta^2_p = .14$. This interaction showed that relative to the CS-, SCRs were significantly greater for the CS+E, $t(146) = 3.50$, $p < .001$, $d = 1.121$, and the CS+U,

**Table 1. SCR results from ANOVAs across experimental phases.**

| Effect | Conditioning Phase (Day 1) DCS *n* = 22, PBO *n* = 21 | | | Extinction Phase (Day 2) DCS *n* = 20, PBO *n* = 18 | | |
|---|---|---|---|---|---|---|
| | Statistic | P-value | $\eta^2_p$ | Statistic | P-value | $\eta^2_p$ |
| Group | $F(1, 41) = 0.30$ | .585 | < .01 | $F(1, 36) = 0.17$ | .687 | < .01 |
| Stimulus | $F(2, 82) = 36.86$ | < .001 | .47 | $F(1, 36) = 2.37$ | .132 | .06 |
| Stage | - | - | - | $F(1, 36) = 21.52$ | < .001 | .37 |
| Group × Stimulus | $F(2, 82) = 0.69$ | .505 | .02 | $F(1, 36) = 1.30$ | .261 | .03 |
| Group × Stage | - | - | - | $F(1, 36) = 1.76$ | .193 | .05 |
| Stimulus × Stage | - | - | - | $F(1, 36) = 1.14$ | .293 | .03 |
| Group × Stimulus × Stage | - | - | - | $F(1, 36) = 3.18$ | .083 | .08 |
| Effect | Recall Phase (Day 3) DCS *n* = 20, PBO *n* = 19 | | | Renewal Phase (Day 3) DCS *n* = 20, PBO *n* = 20 | | |
| | Statistic | P-value | $\eta^2_p$ | Statistic | P-value | $\eta^2_p$ |
| Group | $F(1, 37) = 4.29$ | .553 | < .01 | $F(1, 38) = 0.49$ | .490 | .01 |
| Stimulus | $F(1.39, 51.44) = 4.72$ | .023[a] | .11 | $F(2, 76) = 10.48$ | < .001 | .22 |
| Stage | $F(1, 37) = 31.86$ | < .001 | .46 | $F(1, 38) = 20.73$ | < .001 | .35 |
| Group × Stimulus | $F(1.39, 51.44) = 0.17$ | .761[a] | < .01 | $F(2, 76) = 0.21$ | .812 | < .01 |
| Group × Stage | $F(1, 37) = 1.24$ | .273 | .03 | $F(1, 38) = 0.10$ | .754 | < .01 |
| Stimulus × Stage | $F(1.62, 60.10) = 5.93$ | .007[a] | .14 | $F(2, 76) = 3.70$ | .029 | .09 |
| Group × Stimulus × Stage | $F(1.62, 60.10) = 0.09$ | .872[a] | < .01 | $F(2, 76) = 1.38$ | .258 | .04 |

[a] Greenhouse-Geisser sphericity correction applied.

$t(146) = 4.36$, $p < .001$, $d = 1.395$, at early recall, but not significantly greater in the late stage (both $ps > .619$). The Group × Stimulus and Group × Stimulus × Stage interactions were non-significant, providing no evidence of the hypothesized DCS effects.

**Renewal.** In the renewal phase, there was a significant effect of Stimulus, $F(2, 76) = 10.48$, $p < .001$, $\eta^2_p = .22$, Stage, $F(1, 38) = 20.73$, $p < .001$, $\eta^2_p = .35$, and a Stimulus × Stage interaction $F(2, 76) = 3.70$, $p = .029$, $\eta^2_p = .09$. Post-hoc contrasts indicated that relative to the CS-, SCRs were significantly greater to the CS+E, $t(151) = 2.97$, $p = .004$, $d = 0.937$, and the CS+U, $t(151) = -5.09$, $p < .001$, $d = 1.608$ in the early stage, but not significantly different in the late stage (both $ps > .271$). The Group × Stimulus and Group × Stimulus × Stage interactions were nonsignificant, providing no evidence of the hypothesized DCS effects.

## US expectancy results

Fig 3 shows mean US expectancy ratings for the first and last presentation of each stimulus for all phases of the experiment and Table 2 shows complete statistics of the corresponding ANO-VAs. Significant results are summarized below.

**Conditioning.** There was a significant effect of stimulus on US expectancy at the last presentation of each stimulus during conditioning, $F(2, 80) = 97.8$, $p < .001$, $\eta^2_p = .71$, with significantly greater ratings for the CS+E, $t(80) = -11.49$, $p < .001$, $d = 3.545$, and CS+U, $t(80) = -12.66$, $p < .001$, $d = 3.907$ relative to the CS-. As expected, there were no significant effects of Group or Group × Time.

**Extinction.** In the extinction phase, there were significant main effects of Stimulus, $F(1, 40) = 24.01$, $p < .001$, $\eta^2_p = .38$, and Stage, $F(1, 40) = 75.72$, $p < .001$, $\eta^2_p = .65$, as well as a Stimulus × Stage interaction, $F(1, 40) = 27.54$, $p < .001$, $\eta^2_p = .41$. Post-hoc contrasts showed greater US expectancy ratings for the first presentation of the CS+E relative to the CS-, $t(73.9) = 7.06$,

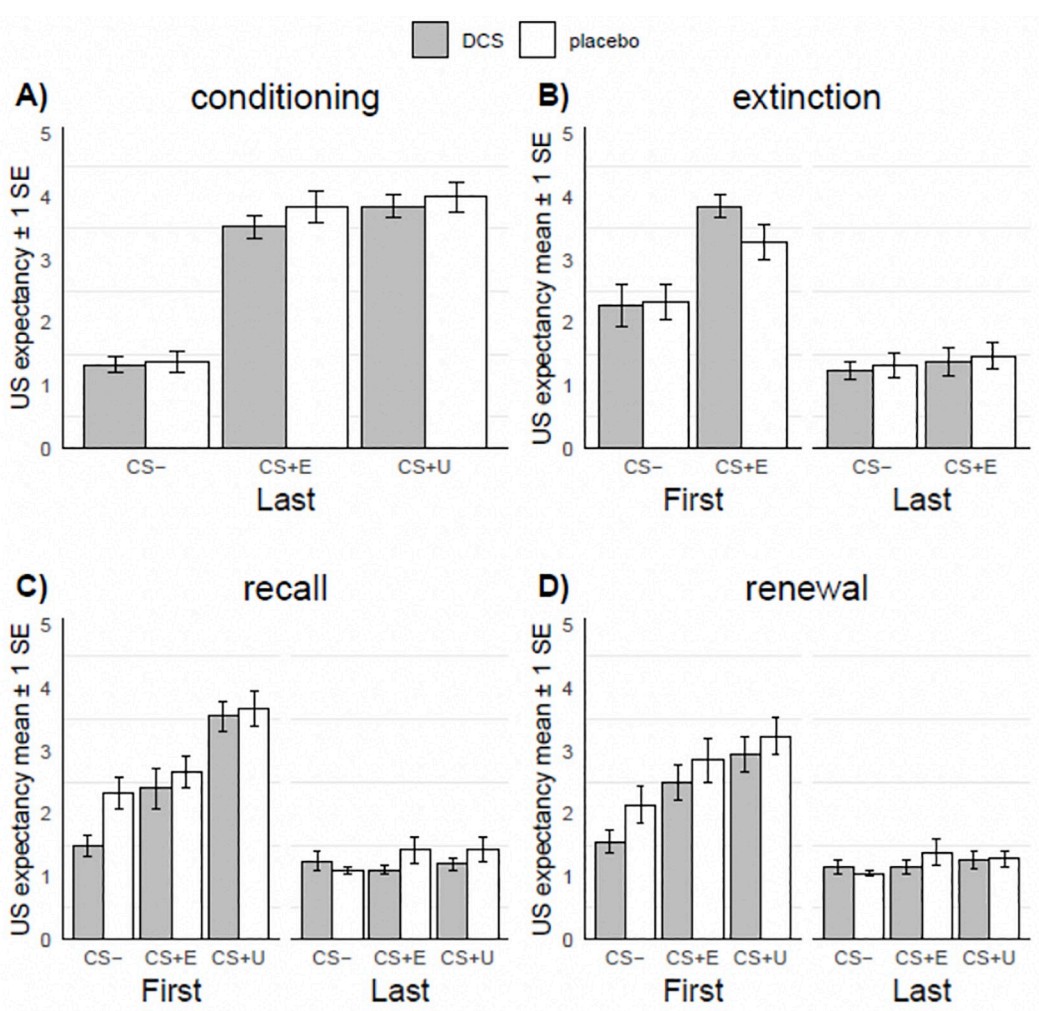

**Fig 3. Expectancy of shock (US expectancy) following first and last presentation of different conditioned stimuli, across experimental phases.** Note. Ratings were made retrospectively (i.e. after phase concluded). PBO = placebo group, DCS = d-cycloserine, SCR = skin conductance response, SE = 1 standard error, CS- = stimulus that was not paired with shock, CS +E = stimulus that was paired with shock during conditioning and presented in the extinction phase, CS+U = stimulus that was paired with shock during conditioning but not presented in the extinction phase.

$p < .001$, $d = 2.180$, but not the last presentation, $t(73.9) = -0.80$, $p = .427$, $d = 0.247$, indicative of successful extinction. All other effects were nonsignificant.

**Recall.** In the recall phase, there were significant main effects of Stimulus, $F(2, 78) = 17.26$, $p < .001$, $\eta^2_p = .31$, and Stage, $F(1, 37) = 197.53$, $p < .001$, $\eta^2_p = .84$, and a significant Stimulus × Stage interaction, $F(2, 78) = 15.53$, $p < .001$, $\eta^2_p = .28$. At the first presentation of each stimulus during recall, US expectancy for the CS+E was significantly greater than the CS-, $t(154) = 2.91$, $p = .004$, $d = 0.910$, indicating a return in expectation of shock since the end of extinction, but significantly less than the CS+U, $t(154) = 5.08$, $p < .001$, $d = 1.587$. Differences across stimuli were non-significant ($p > .504$) for the last presentation of each stimulus. There was a significant main effect of Group, $F(1, 39) = 4.42$, $p = .042$, $\eta^2_p = .10$, indicating lower US expectancyratings across all stimuli and stages in the DCS group relative to PBOgroup. However, all interactions with Group were nonsignificant (all $ps > .096$).

**Renewal.** In the renewal phase, there was a significant main effect of Stimulus, $F(2, 76) = 11.11$, $p < .001$, $\eta^2_p = .23$, and Stage, $F(1, 38) = 98.27$, $p < .001$, $\eta^2_p = .72$, and a significant

**Table 2. US expectancy results from ANOVAs across experimental phases.**

| Effect | Conditioning Phase (Day 1) DCS $n = 21$, PBO $n = 21$ | | | Extinction Phase (Day 2) DCS $n = 21$, PBO $n = 21$ | | |
|---|---|---|---|---|---|---|
| | Statistic | P-value | $\eta^2_p$ | Statistic | P-value | $\eta^2_p$ |
| Group | $F(1, 40) = 1.34$ | .254 | .03 | $F(1, 40) = 0.16$ | .696 | < .01 |
| Stimulus | $F(2, 80) = 97.88$ | < .001 | .71 | $F(1, 40) = 24.01$ | < .001 | .38 |
| Stage | - | - | - | $F(1, 40) = 75.72$ | < .001 | .65 |
| Group × Stimulus | $F(2, 80) = 0.26$ | .774 | < .01 | $F(1, 40) = 1.17$ | .287 | .03 |
| Group × Stage | - | - | - | $F(1, 40) = 0.96$ | .332 | .02 |
| Stimulus × Stage | - | - | - | $F(1, 40) = 27.54$ | < .001 | .41 |
| Group × Stimulus × Stage | - | - | - | $F(1, 40) = 2.11$ | .154 | .05 |
| | Recall Phase (Day 3) DCS $n = 20$, PBO $n = 21$ | | | Renewal Phase (Day 3) DCS $n = 20$, PBO $n = 20$ | | |
| Effect | Statistic | P-value | $\eta^2_p$ | Statistic | P-value | $\eta^2_p$ |
| Group | $F(1, 39) = 4.42$ | .042 | .10 | $F(1, 38) = 1.57$ | .217 | .04 |
| Stimulus | $F(2, 78) = 17.26$ | < .001 | .31 | $F(2, 76) = 11.11$ | < .001 | .23 |
| Stage | $F(1, 37) = 197.53$ | < .001 | .84 | $F(1, 38) = 98.27$ | < .001 | .72 |
| Group × Stimulus | $F(2, 78) = 0.15$ | .859 | < .01 | $F(2, 76) = 0.05$ | .951 | < .01 |
| Group × Stage | $F(1, 37) = 1.76$ | .192 | .04 | $F(1, 38) = 1.51$ | .227 | .04 |
| Stimulus × Stage | $F(2, 78) = 15.53$ | < .001 | .28 | $F(2, 76) = 8.78$ | < .001 | .19 |
| Group × Stimulus × Stage | $F(2, 78) = 2.41$ | .096 | .06 | $F(2, 76) = 0.75$ | .476 | .02 |

Stimulus × Stage interaction, $F(2, 76) = 8.78$, $p < .001$, $\eta^2_p = .19$. At the first presentation of each stimulus during renewal, US expectancy was significantly greater for the CS+E relative to the CS-, $t(147) = 4.28$, $p < .001$, $d = 1.352$, and for the CS+U relative to the CS-, $t(147) = 6.14$, $p < .001$, $d = 1.943$, but not significantly different between the CS+E and CS+U, $p = .064$. Differences across stimuli were non-significant ($p > .39$) for the last presentation of each stimulus. There were no significant group effects or interactions.

## Discussion

The aim of this study was to test whether DCS facilitates fear extinction retention in patients with social anxiety disorder. To this end, we employed a paradigm developed to study fear extinction and that mirrors that used in the study of fear extinction in rodents. We observed several noteworthy findings. First, of the 81 participants that we enrolled in this study, only 43 (53%) provided data that we could use for the analyses. Specifically, we had to remove 12 participants because their SCR recordings that were consistent with data collection errors and an additional 25 participants because they failed to show fear acquisition. Such acquisition failure rates are not uncommon in human fear conditioning studies [23,24], and may have resulted from a relatively low reinforcement rate used during conditioning [31], or because clinical populations are less likely to demonstrate differential conditioning [37], even when 100% reinforcement schedules are used [24]. Accordingly, our acquisition results are well in line with expectations from the literature. A necessary consequence to our decision to examine extinction effects only in those who had acquired a differential response is that our results are necessarily specific to individuals who learned a conditioned fear. Nonetheless, there were no differences in clinical severity, contingency awareness, or US expectancy between conditioners and non-conditioners. Moreover, results did not differ when non-conditioners were included in the analysis.

Second, despite selection of those displaying adequate fear acquisition, fear retention (and a stimulus by phase interaction) at the outset of the Day Two extinction phase was evident only for the expectancy measure not for SCR. This flattening of the differential responding between

the CS+ and CS- may reflect a combination of stimulus generalization and poor consolidation, although it is clear from the recall and renewal effects that greater fear learning to the CS+ persisted relative to the CS-. In addition, extinction of reactivity to both CSs was achieved across phases, presumably providing adequate extinction learning for augmentation.

Third, under these conditions, we found no evidence for the hypothesis that preceding extinction training with a single dose of 50 mg of DCS would enhance fear extinction retention. Aside from failing to demonstrate that the observed clinical effects of DCS for augmenting exposure therapy may indeed be explained by engaging the core hypothesized mechanism—i.e., fear extinction retention—this study shows that demonstrating fear extinction in patients using this specific paradigm presents major challenges.

This study is an important addition to the literature on DCS for enhancing extinction learning. We are aware of two other groups of studies that have failed to find a significant advantage for DCS for augmenting extinction training in the laboratory [17,18]. Yet, across these studies there were a number of methodological issues that may have hindered the efficacy of DCS augmentation. First, in the Guastella et al. [18] series of studies, the first two studies conducted acquisition and extinction procedures on the same day, separated only by a few hours and with DCS administration given in the interval between procedures. As such, as acknowledged by the authors, DCS was given well within the consolidation window of acquisition and hence could have had facilitative effects on both acquisition and extinction. This design flaw was corrected in the third study, but in all three studies conducted by Guastella et al. [18] participants were asked to record, using a rotary dial rating, the subjective expectancy of shock during each CS presentation. Such procedures are assumed to enhance explicit, higher order (e.g., propositional) processing of the fear contingency [38]. Notably, in a review of early DCS studies, Grillon [39] hypothesized that DCS benefits may be specific to lower-order learning processes. Specifically, Grillon discussed a dual-model theory of fear conditioning, where human de novo fear conditioning processes may rely on higher-order cognition and that DCS benefits may be specific to lower-order, associative processes more characteristic of animal paradigms and human clinical fears. Interestingly, concerns about the influence of higher order processes are also apt for the DCS conditioning study by Klumpers and colleagues [17], where, after the first block of acquisition, participants were informed of the CS-shock association—thereby helping ensure higher order encoding of the causal relationships.

In the current study, we reduced cues for higher order processing, while retaining some ratings of explicit knowledge of the fear contingency in the form of retrospective expectancy ratings and questions about which colors were followed by shock, administered at the end of each phase. Under these conditions, we failed to find a DCS augmentation effect. In light of previous null findings of DCS on extinction retention, it appears unclear whether DCS can enhance extinction recall or reduce renewal in a human de novo conditioning paradigm. Given the limitations of the present study (e.g. relatively weak conditioning as measured by SCR) and evidence that DCS can reduce reinstatement [19,20], it would be worth further investigating the procedural variants that might enable detection of DCS augmentation effects found in animal and human clinical research (see [40] for a discussion of needed improvements to human fear conditioning paradigms). For instance, using biologically "prepared" or other fear-relevant stimuli can lead to stronger conditioned responses that reflect a greater role of lower-order fear learning processes [41,42], and therefore may be more susceptible to the effects of DCS.

## Supporting information

**S1 File. Dataset.** This is the complete dataset used for analyses.
(CSV)

**S2 File. Supplementary analyses.** Sensitivity analyses were run with all available data to determine whether exclusion of non-conditioners affected the impact of DCS vs. PBO on extinction retention. Consistent with results when analyzing only conditioners, no main or interactive effects of group were seen during recall or renewal phases for SCR or US expectancy data. (HTML)

## Author Contributions

**Conceptualization:** Stefan G. Hofmann, Mark H. Pollack, Jasper A. J. Smits.

**Data curation:** Santiago Papini, Michael W. Otto, David Rosenfield, Mara Lewis, Sara Witcraft, Mark H. Pollack, Jasper A. J. Smits.

**Formal analysis:** Santiago Papini, David Rosenfield.

**Funding acquisition:** Stefan G. Hofmann, David Rosenfield, Mark H. Pollack, Jasper A. J. Smits.

**Investigation:** Stefan G. Hofmann, Michael W. Otto, Sheila Dowd, Mark H. Pollack, Jasper A. J. Smits.

**Methodology:** Stefan G. Hofmann, Santiago Papini, Michael W. Otto, David Rosenfield, Sheila Dowd, Mark H. Pollack, Jasper A. J. Smits.

**Project administration:** Joseph K. Carpenter, Sheila Dowd, Mara Lewis, Sara Witcraft.

**Software:** Joseph K. Carpenter, Christina D. Dutcher, Sheila Dowd.

**Supervision:** Stefan G. Hofmann, Jasper A. J. Smits.

**Validation:** Joseph K. Carpenter, Christina D. Dutcher, Sara Witcraft.

**Visualization:** Santiago Papini, Joseph K. Carpenter, Christina D. Dutcher.

**Writing – original draft:** Stefan G. Hofmann, Mark H. Pollack.

**Writing – review & editing:** Stefan G. Hofmann, Santiago Papini, Joseph K. Carpenter, Michael W. Otto, David Rosenfield, Christina D. Dutcher, Jasper A. J. Smits.

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
