## [Decision Letter · Decision Letter 0]

16 Jul 2019

PONE-D-19-15283

Effect of d-cycloserine on fear extinction training in adults with social anxiety disorder

PLOS ONE

Dear Dr. Hofmann,

Thank you for submitting your manuscript to PLOS ONE. After careful consideration, we feel that it has merit but does not fully meet PLOS ONE’s publication criteria as it currently stands. Therefore, we invite you to submit a revised version of the manuscript that addresses the points raised during the review process. Please find the reviewer comments below.

We would appreciate receiving your revised manuscript by Aug 30 2019 11:59PM. To enhance the reproducibility of your results, we recommend that if applicable you deposit your laboratory protocols in protocols.io, where a protocol can be assigned its own identifier (DOI) such that it can be cited independently in the future. For instructions see: http://journals.plos.org/plosone/s/submission-guidelines#loc-laboratory-protocols

We look forward to receiving your revised manuscript.

Kind regards,

Judith Homberg

Academic Editor

PLOS ONE

Journal Requirements:

Reviewers' comments:

Reviewer's Responses to Questions

**Comments to the Author**

1. Is the manuscript technically sound, and do the data support the conclusions?

Reviewer #1: Yes

Reviewer #2: Partly

2. Has the statistical analysis been performed appropriately and rigorously? 

Reviewer #1: No

Reviewer #2: Yes

3. Have the authors made all data underlying the findings in their manuscript fully available?

Reviewer #1: Yes

Reviewer #2: Yes

4. Is the manuscript presented in an intelligible fashion and written in standard English?

Reviewer #1: Yes

Reviewer #2: Yes

5. Review Comments to the Author

Reviewer #1: The manuscript ‘Effect of d-cycloserine on fear extinction training in adults with social anxiety disorders’ describes the results of a three-day fear learning paradigm. Participants were randomly allocated to receive 50 mg d-cycloserine (DCS) or placebo one hour prior to extinction training on day 2. Data of 43 participants suggested that the fear-learning task successfully produced the desired learning effects, but that DCS did not moderate any effects. The authors conclude that they found no evidence for the putative mechanism of action of DCS: enhancement of extinction memory consolidation.

This study should be considered an important contribution to the literature. Thus far, no study has investigated whether DCS enhances extinction memory consolidation during a de novo fear conditioning paradigm in a clinical population. As such, the current study fills an important gap in the translation of pre-clinical work in healthy controls to treatment interventions for those suffering from anxiety disorders. Moreover, the current study makes use of a three-day paradigm, allowing to disentangle the learning and memory effects of acquisition, extinction and retention. However, the current manuscript suffers also from some weaknesses, which should be addressed.

Abstract:

• The authors state that “human studies of DCS augmentation in a de novo fear paradigm have been scarce and inconclusive”. However, in the introduction they report that all these studies had null-findings. The authors may want to rephrase their summary of findings in the abstract.

• The findings of the study are summarized in two sentences in the abstract. The authors should consider discussing their findings in greater detail and formulating their findings related to the experimental phases (and thus the hypotheses), instead of merely stating that DCS did not moderate fear responses.

Introduction:

• On page 4. the authors write: “Despite this hope, and despite the wealth of clinical trial data showing DCS augmentation success…”. Clinical trial data has shown both DCS augmentation success and failure. A more balanced statement would better reflect the overall clinical trial data for DCS augmentation.

• Was the decision to only include those who demonstrated adequate conditioning of de novo fears an a-priori or post-hoc decision? Of note, performance-based exclusion is not always recommended (see f.i. Lonsdorf et al., 2017 Don’t fear fear conditioning. Neuroscience and Biobehav Reviews). The authors should consider performing additional analyses on all available data.

• On line 85/86 page 4, the authors state that the aims, hypotheses, design, and planned statistical analyses of this experiment were published. However, the protocol paper describes the aim, design and outcome of interest, but these do not completely overlap with the current report. The authors have done different analyses than they originally planned and should explain why they changed their plans.

• The authors should formulate their hypotheses in terms of the experimental phases: What were the specific hypotheses regarding extinction recall and renewal?

Methods:

• The participants self-selected to participate in the experiment. Less than half of participants in the clinical trial chose to participate in the experiment. Is there any information available regarding reasons to not participate in the current study?

• What was the reason for choosing a reinforcement rate of 62.5%? The authors should comment on that in the method section of the manuscript. (In addition, could this low reinforcement rate be related to the failure to acquire fear in half of the sample? The authors should critically discuss this in their discussion section).

• On page 8, line 166: should safe context be danger context?

• Was the decision to analyze US expectancy ratings as a secondary outcome made a-priori or post-hoc? Please clarify.

• In the description of the statistical analyses for the recall and renewal phases the “phase term" seems to be missing.

Results:

• Did those who did not demonstrate discriminant SCR conditioning also not demonstrate explicit contingency learning as indexed by US expectancies? Did those measures align? If not, what was the overlap between measures?

• The sample sizes in table 1 are a bit confusing: Why are there differences in sample size between phases? Please also address this in the Table notes.

• For the SCR, the stimulus by stage interaction is not significant in the extinction phase. How should this be interpreted? Please clarify.

• Page 14, line 284-285. This sentence is slightly confusing, please consider rewriting.

Discussion:

• The decision to exclude those that showed no differential SCR conditioning should be critically discussed (see Marin et al., 2019 Absence of conditioned responding in humans: A bad measure or individual differences; Lonsdorf et al., 2017)

• The discussion should include a critical reflection on the experimental design (e.g. the reinforcement rate, the outcome measures, the stimuli used (CS’s and US), etc.)

• The authors should consider including future research directions in their discussion.

Reviewer #2: The authors studied the effect of a single dose of the N-methyl-D-aspartate partial agonist D-cycloserine (DCS; 50mg) on fear extinction in patients with diagnosed social anxiety disorder using a de novo fear conditioning and extinction paradigm in the laboratory over 3 experimental days. It was found that pre-extinction DCS did not affect physiological (skin conductance response (SCR)) and psychological (shock expectancy) parameters assessed during fear extinction training, recall or renewal. The authors conclude that DCS has no benefit in humans in de novo fear extinction.

The present data adds on numerous studies showing variable effects of DCS on fear extinction and exposure based therapy in healthy and psychopathological subjects (humans and animals), respectively. I have a number of concerns regarding the design of the study, the presentation and interpretation of the results as outlined below in detail:

Major concerns:

1. Several important procedural and technical details are missing and need to be added before a final sound evaluation of this manuscript can be made. Authors should refer e.g. to table 6 in Lonsdorf et al 2017, Neuroscience and Biobehavioral reviews to provide missing information, such as for example:

a. Please provide details on the skin conductance measure. Which system and which settings were used? Which sampling rate?

b. Please provide the data (mean + SD) of actually used shock intensities of the unconditioned stimulus on pp6.

c. Please explain why (different to previous papers of the group) this time partial contingency was used, pairing 5 out of 8 CS with the US.

d. Indicate what is considered as early and late stages of the diverse extinction sessions.

2. The main conclusion of the authors that DCS does not affect fear extinction in this experimental setup cannot be drawn from the presented data. In my opinion the data do not support the formation of a fear memory that could be extinguished in the study population. First, no usable SCR was obtained for 15% of the participants due to technical problems and 38% of the remaining participants did not learn the fear association adequately (as indicated by no difference in SCR to CS- and CS+) and, thus, were excluded from the analysis. The fact that only half of the study population was used, raises questions whether this paradigm was suited for social anxiety patients. Importantly, placebo controls showed the same SCR in response to the CS+ as to the CS- during early extinction phase suggesting that no fear memory was formed. Furthermore, there seems to be no fear extinction, as the reduction in SCR is similarly present in unconditioned and conditioned placebo controls suggesting habituation rather than fear extinction. Finally, in contrast to the (early) extinction trial, CS+ groups showed higher fear responses than CS- groups in extinction recall and renewal sessions. This could be an effect of fear reconsolidation. Please explain.

3. Several recent meta-analysis and publications (including important contributions some of the authors of the present MS) critically discuss the importance of timepoint of DCS administration when used as an adjunct to exposure based therapy. Against the recommendation that DCS should be administered AFTER successful fear extinction training (i.e. reduction in fear responses) in order to reduce the risk of enhancing the previously formed fear memory by facilitating reconsolidation, the authors administered DCS PRIOR to fear extinction training. Please explain and critically discuss why this timepoint for DCS administration was chosen.

4. Figure legends 2 and 3 are sloppy, they just give a title and do not provide information on the data presented and the abbreviations in the figures (kind of data shown (mean +/- sem?). Needs to be revised.

Minor points:

5. Figure 1. For a better temporal resolution of the experimental protocol, a timeline including the images presented in figure 1 and procedures (pairings details also given below the images) should be included. Please also show habituation period in the timeline.

6. In general, some parts of the MS are written in present tense rather than past tense. Please revise.

7. On pp8, line 159 check description of the safe context. Furthermore, on the same page, on line 166 I think the contexts for renewal got mixed up and it should be “within threat context” rather than “safe” context. Please correct.

8. Please clarify the abbreviation PBO mentioned for the first time on pp9, line 194. Please either stick to PBO or placebo throughout the manuscript.

9. Table 1 and 2 seem to be redundancies of the main text (pp11-15).

6. PLOS authors have the option to publish the peer review history of their article (what does this mean?). If published, this will include your full peer review and any attached files.

Reviewer #1: No

Reviewer #2: No

---

## [Author Response · Author response to Decision Letter 0]

27 Aug 2019

Reviewer #1: The manuscript ‘Effect of d-cycloserine on fear extinction training in adults with social anxiety disorders’ describes the results of a three-day fear learning paradigm. Participants were randomly allocated to receive 50 mg d-cycloserine (DCS) or placebo one hour prior to extinction training on day 2. Data of 43 participants suggested that the fear-learning task successfully produced the desired learning effects, but that DCS did not moderate any effects. The authors conclude that they found no evidence for the putative mechanism of action of DCS: enhancement of extinction memory consolidation.

This study should be considered an important contribution to the literature. Thus far, no study has investigated whether DCS enhances extinction memory consolidation during a de novo fear conditioning paradigm in a clinical population. As such, the current study fills an important gap in the translation of pre-clinical work in healthy controls to treatment interventions for those suffering from anxiety disorders. Moreover, the current study makes use of a three-day paradigm, allowing to disentangle the learning and memory effects of acquisition, extinction and retention. However, the current manuscript suffers also from some weaknesses, which should be addressed.

Abstract:

• The authors state that “human studies of DCS augmentation in a de novo fear paradigm have been scarce and inconclusive”. However, in the introduction they report that all these studies had null-findings. The authors may want to rephrase their summary of findings in the abstract.

Response: We have revised the respective sentences in order to make them more concordant, as seen below.

Abstract:

“Preclinical and clinical data have shown that D-cycloserine (DCS), a partial agonist at the N-methyl-d-aspartate receptor complex, augments the retention of fear extinction in animals and the therapeutic learning from exposure therapy in humans. However, studies with non-clinical human samples in de novo fear conditioning paradigms have demonstrated minimal to no benefit of DCS.”

Page 4:

“Despite this hope, and despite evidence of successful DCS augmentation in clinical trials, initial human studies using de novo fear conditioning paradigms have shown minimal to no effect of DCS on the retention of extinction learning. Specifically, studies examining the effects of DCS augmentation on extinction recall and fear renewal have consistently had null findings [15–18], though some evidence of reduced reinstatement has been found [17,18].”

• The findings of the study are summarized in two sentences in the abstract. The authors should consider discussing their findings in greater detail and formulating their findings related to the experimental phases (and thus the hypotheses), instead of merely stating that DCS did not moderate fear responses.

Response: We have revised the abstract accordingly:

“The primary outcome was skin conductance response to conditioned stimuli, and shock expectancy ratings were examined as a secondary outcome. Results showed greater skin conductance and expectancy ratings in response to the CS+ compared to CS- at the end of conditioning. As expected, this difference was no longer present at the end of extinction training, but returned at early recall and renewal phases on Day 3, showing evidence of return of fear. In contrast to hypotheses, DCS had no moderating influence on skin conductance response or expectancy of shock during recall or renewal phases.

Introduction:

• On page 4. the authors write: “Despite this hope, and despite the wealth of clinical trial data showing DCS augmentation success…”. Clinical trial data has shown both DCS augmentation success and failure. A more balanced statement would better reflect the overall clinical trial data for DCS augmentation.

Response: We have ensured our statements of DCS’s clinical effects are balanced with the following on page 3:

Although results across individual clinical trials have been variable [e.g., 7,8], a recent meta-analysis indicates that across disparate clinical trials of anxiety disorders, DCS augmentation of exposure therapy offers advantages on the order of a small effect size (d = 0.25) for enhancing early response to treatment relative to placebo [9].

And page 4:

Despite this hope, and despite evidence of successful DCS augmentation in clinical trials...

• Was the decision to only include those who demonstrated adequate conditioning of de novo fears an a-priori or post-hoc decision? Of note, performance-based exclusion is not always recommended (see f.i. Lonsdorf et al., 2017 Don’t fear fear conditioning. Neuroscience and Biobehav Reviews). The authors should consider performing additional analyses on all available data.

Response: We decided a priori to exclude non-conditioners, as this is a consistent approach for our team members. We have also provided additional clarification for our rationale, including citations to our past studies and an additional citation showing that conditioning failures reflect differential brain processing of the conditioning stimuli. 

Page 4-5: “Third, to provide a direct test of extinction effects that have an analogue to clinical fears, we assessed DCS vs. placebo augmentation effects only in individuals who demonstrated adequate acquisition of de novo fears (indeed, among both anxious and healthy samples a substantial proportion of participants may fail to show fear acquisition on skin conductance measures [18,23,24], and there is evidence that poor skin conductance conditioning reflects hypoactivation of brain regions involved in fear learning and expression [25].”

And on page 10: “This approach is consistent with our previous work [21,22], and was done to ensure that participants included in the analysis demonstrated adequate fear learning that could meaningfully be subjected to extinction and renewal procedures in the subsequent phases of the study (see also Marin et al. 2019).”

Yet, we are aware that different opinions exist on this methodological strategy, and to respond to such concerns, we include the following on pages 11-12

“Following the recommendations of Lonsdorf et al., 2019, we performed sensitivity analyses to determine whether exclusion of non-conditioners influences results. No differential effects were obtained relative to those reported below, and we report these results as supplementary material.”

• On line 85/86 page 4, the authors state that the aims, hypotheses, design, and planned statistical analyses of this experiment were published. However, the protocol paper describes the aim, design and outcome of interest, but these do not completely overlap with the current report. The authors have done different analyses than they originally planned and should explain why they changed their plans.

We have clarified this on pg. 5: “Prior to data analysis we made several modifications to the analytic approach described in Hofmann et al. (2015) [23] to be consistent with the latest methodological advancements and recommendations. Specifically, we used continuous decomposition analysis to extract skin conductance responses and we tested the pre-specified hypotheses in ANOVA that included a term for contrasts between stimuli, as opposed to subtracting CS- SCRs from CS+ SCRs prior to analyses [27]. Another modification was to omit the prespecified “Extinction Retention Index” (ERI) analysis in light of a recent publication [28] which outlined theoretical and procedural problems with its operationalization, including the existence of 16 different calculations of the ERI in the literature.”

The authors should formulate their hypotheses in terms of the experimental phases: What were the specific hypotheses regarding extinction recall and renewal?

We have done so on pg. 5: 

“Prior to study initiation, hypotheses (i.e., DCS enhancement of extinction recall and reduction of fear renewal) were published in Hofmann et al. [26].”

Methods:

• The participants self-selected to participate in the experiment. Less than half of participants in the clinical trial chose to participate in the experiment. Is there any information available regarding reasons to not participate in the current study?

Response: This specific experiment was optional and considered a separate study from the clinical trial. All eligible participants were given the option to participate. We did not collect information regarding the reasons for or against choosing to participate.

• What was the reason for choosing a reinforcement rate of 62.5%? The authors should comment on that in the method section of the manuscript. (In addition, could this low reinforcement rate be related to the failure to acquire fear in half of the sample? The authors should critically discuss this in their discussion section).

Response: We now write on page 9:

“This reinforcement rate was used to replicate procedures from the previously validated paradigm used for this study [21,22], and because lower reinforcement rates create more uncertainty and lead to slower extinction [31,32], thus allowing more room for potential DCS augmentation effects.” 

And page 17 of the discussion:

Such acquisition failure rates are not uncommon in human fear conditioning studies [23,24], and may have resulted from a relatively low reinforcement rate used during conditioning [32], or because clinical populations are less likely to demonstrate differential conditioning [38], even when 100% reinforcement schedules are used [24].

• On page 8, line 166: should safe context be danger context?

Response: We have corrected this to reflect the context used in the renewal phase.

• Was the decision to analyze US expectancy ratings as a secondary outcome made a-priori or post-hoc? Please clarify.

We now specify that our primary hypothesis relates to SCR on page 5, which is consistent with the published protocol paper (Hofmann et al., 2015).

“Our primary hypothesis was that DCS would augment de novo fear extinction learning of SCR through increased retention of extinction during a recall and renewal phase occurring 24 hours later.”

We clarify that the decision to analyze expectancy ratings as a secondary outcome was made post-hoc on page 4:

“Second, we reduced cues for higher-order processing (i.e. shock expectancy ratings administered during CS presentations), while retaining some assessment of explicit knowledge of the fear contingency in the form of retrospective expectancy ratings administered at the end of each experimental phase. This enabled us to evaluate post-hoc whether DCS effects on skin conductance response (SCR) were mirrored by expectancy ratings.” 

• In the description of the statistical analyses for the recall and renewal phases the “phase term" seems to be missing.

Response: We have added the term into the description (pg. 11)

Results:

• Did those who did not demonstrate discriminant SCR conditioning also not demonstrate explicit contingency learning as indexed by US expectancies? Did those measures align? If not, what was the overlap between measures?

Response: We write on page 11:

“Total LSAS score,... MADRS score,…and US intensity... were not significantly different between participants that did and did not show differential SCR conditioning, nor was differential US expectancy (CS+E minus CS-), t(66) = -0.59, p = 0.56, conditioners: M = 2.33, SD = 1.34; non-conditioners: M = 2.11, SD = 1.70, or likelihood of contingency awareness, χ2 (1) = 2.42, p = 0.120, at the end of the conditioning phase.”

• The sample sizes in table 1 are a bit confusing: Why are there differences in sample size between phases? Please also address this in the Table notes.

Response: We have clarified this in the methods section (pg. 10): 

“Since hypotheses were tested within each phase, participants were not excluded from analysis in one phase when they had incomplete data in another phase, which resulted in minor variations in sample size across phases.”

• For the SCR, the stimulus by stage interaction is not significant in the extinction phase. How should this be interpreted? Please clarify.

Response: Our SCR data did reflect a lack of specificity for the extinction effects observed on Day 2: there was a decrease in responding across the CSs, without the stated interaction reaching significance. The most reasonable hypotheses for the failure of this interaction effect is generalization occurring between Day 1 and Day 2, so that safety was learned across cues that shared at least some fear associations (due to shared context, heightened shock expectation for all stimuli,or diminished retention of “fear” of the CS+). Nonetheless, results for the expectancy ratings were clear (i.e., had the expected stimulus by stage interaction), and differential conditioning for SCR was evident on Day 3 in the recall and renewal paradigms, supporting the notion that learning persisted to Day 3. Finally, despite the lack of clear differential SCR at extinction initiation, reactivity to both CS+ and CS- decreased significantly across the extinction phase, presumably providing an extinction learning substrate for DCS augmentation. In the revised discussion we now devote a paragraph to explicating these issues (page 17-18):

“Second, despite selection of those displaying adequate fear acquisition, fear retention (and a stimulus by phase interaction) at the outset of the Day Two extinction phase was evident only for the expectancy measure not for SCR. This flattening of the differential responding between the CS+ and CS- may reflect a combination of stimulus generalization and poor consolidation, although it is clear from the recall and renewal effects that greater fear learning to the CS+ persisted relative to the CS-. In addition, extinction of reactivity to both CSs was achieved across phases, presumably providing adequate extinction learning for augmentation.”

• Page 14, line 284-285. This sentence is slightly confusing, please consider rewriting.

We have revised the sentence as following (now on page 16):

“There was a significant main effect of Group, F(1, 39) = 4.42, p = .042, η²p = .10, indicating lower US expectancy ratings across all stimuli and stages in the DCS group relative to PBO. However, all interactions with Group were nonsignificant (all ps > .096).” 

Discussion:

• The decision to exclude those that showed no differential SCR conditioning should be critically discussed (see Marin et al., 2019 Absence of conditioned responding in humans: A bad measure or individual differences; Lonsdorf et al., 2017)

Response: As described in response to the point above, we have now provided, on pages 5 and 10 a more complete rationale for our exclusion of those showing no differential conditioning, including citations of both Marin and Londsdorf in that section. 

We also write on page 17-18:

“Such acquisition failure rates are not uncommon in human fear conditioning studies [23,24], and may have resulted from a relatively low reinforcement rate used during conditioning [32], or because clinical populations are less likely to demonstrate differential conditioning [38], even when 100% reinforcement schedules are used [24]. Accordingly, our acquisition results are well in line with expectations from the literature. A necessary consequence to our decision to examine extinction effects only in those who had acquired a differential response is that our results are necessarily specific to individuals who learned a conditioned fear. Nonetheless, there were no differences in clinical severity, contingency awareness, or US expectancy between conditioners and non-conditioners. Moreover, results did not differ when non-conditioners were included in the analysis.”

• The discussion should include a critical reflection on the experimental design (e.g. the reinforcement rate, the outcome measures, the stimuli used (CS’s and US), etc.)

• The authors should consider including future research directions in their discussion.

Response: In addition to our response to the preceding point, we have written the following on page 19-20 of the discussion.

“Given the limitations of the present study (e.g. relatively weak conditioning as measured by SCR) and evidence that DCS can reduce reinstatement [19,20], it would be worth further investigating the procedural variants that might enable detection of DCS augmentation effects found in animal and human clinical research (see [41] for a discussion of needed improvements to human fear conditioning paradigms). For instance, using biologically “prepared” or other fear-relevant stimuli can lead to stronger conditioned responses that reflect a greater role of lower-order fear learning processes [42,43], and therefore may be more susceptible to the effects of DCS.

Reviewer #2: The authors studied the effect of a single dose of the N-methyl-D-aspartate partial agonist D-cycloserine (DCS; 50mg) on fear extinction in patients with diagnosed social anxiety disorder using a de novo fear conditioning and extinction paradigm in the laboratory over 3 experimental days. It was found that pre-extinction DCS did not affect physiological (skin conductance response (SCR)) and psychological (shock expectancy) parameters assessed during fear extinction training, recall or renewal. The authors conclude that DCS has no benefit in humans in de novo fear extinction.

The present data adds on numerous studies showing variable effects of DCS on fear extinction and exposure based therapy in healthy and psychopathological subjects (humans and animals), respectively. I have a number of concerns regarding the design of the study, the presentation and interpretation of the results as outlined below in detail:

Major concerns:

1. Several important procedural and technical details are missing and need to be added before a final sound evaluation of this manuscript can be made. Authors should refer e.g. to table 6 in Lonsdorf et al 2017, Neuroscience and Biobehavioral reviews to provide missing information, such as for example:

a. Please provide details on the skin conductance measure. Which system and which settings were used? Which sampling rate?

We now write on page 7:

“At the University of Texas-Austin site, a BIOPAC MP150 Psychophysiological Recording Apparatus (BIOPAC Systems, Inc., USA), was used, and data were acquired using AcqKnowledge 4.0 software. At Boston University and Rush University, psychophysiological data were recorded with custom equipment made by James Long Company, Caroga Lake, NY, and the data-acquisition program Snap-Master for Windows. Across sites, the sampling rate was 1000 Hz.”

b. Please provide the data (mean + SD) of actually used shock intensities of the unconditioned stimulus on pp6.

Response: This has been added on page 7:

“...mean shock intensity was 1.86 Milliamperes (SD = 1.56).”

We also added a comparison of shock intensity between conditioners and non-conditioners to check whether this variable may have been related to likelihood of skin conductance conditioning, which it was not.

Page 11: “...US intensity (i.e. individually selected shock level), t(64) = -.35, p = .725 [was] not significantly different between participants that did and did not show differential SCR conditioning…”

c. Please explain why (different to previous papers of the group) this time partial contingency was used, pairing 5 out of 8 CS with the US.

Response: The reinforcement rate described in the protocol paper (Hofmann et al., 2015) was in error, as the original protocol as described in the NIH grant funding this study stated that a partial reinforcement rate of 62.5% (5 of 8 CS+ presentations) would be used. 

d. Indicate what is considered as early and late stages of the diverse extinction sessions.

We write on page 10:

“(6) for the conditioning phase, the last four SCRs of each stimulus were averaged to calculate late conditioning, and for the remaining phases, the first two (early) and last two (late) trials were averaged;” 

2. The main conclusion of the authors that DCS does not affect fear extinction in this experimental setup cannot be drawn from the presented data. In my opinion the data do not support the formation of a fear memory that could be extinguished in the study population. First, no usable SCR was obtained for 15% of the participants due to technical problems and 38% of the remaining participants did not learn the fear association adequately (as indicated by no difference in SCR to CS- and CS+) and, thus, were excluded from the analysis. The fact that only half of the study population was used, raises questions whether this paradigm was suited for social anxiety patients. Importantly, placebo controls showed the same SCR in response to the CS+ as to the CS- during early extinction phase suggesting that no fear memory was formed. Furthermore, there seems to be no fear extinction, as the reduction in SCR is similarly present in unconditioned and conditioned placebo controls suggesting habituation rather than fear extinction. Finally, in contrast to the (early) extinction trial, CS+ groups showed higher fear responses than CS- groups in extinction recall and renewal sessions. This could be an effect of fear reconsolidation. Please explain.

Is this actually significant?

Response: In the revision we better explain that our rates of successful/unsuccessful acquisition are in line with results in the literature - this is simply a surprisingly/ concerningly common result in the fear conditioning literature. Also, as explicated in paragraph 2 of our discussion, we are able to track the differential conditioning to the CS+ across the three days of our study (and note that it is evident in the expectancy ratings when it is not evident in SCR). Accordingly, we do believe we have a sample with adequate learning to have the potential to demonstrate a DCS effect. Yet we did not, a result that is also in line with the extant literature. In the discussion we now more clearly present the case for an adequate acquisition and extinction substrate for a fair DCS vs. Placebo experiment. Moreover, we now present results for the full sample in addition to our highly relevant conditioners. Given these factors, we do believe that our results are noteworthy (significant) and are a contribution to the field. 

3. Several recent meta-analysis and publications (including important contributions some of the authors of the present MS) critically discuss the importance of timepoint of DCS administration when used as an adjunct to exposure based therapy. Against the recommendation that DCS should be administered AFTER successful fear extinction training (i.e. reduction in fear responses) in order to reduce the risk of enhancing the previously formed fear memory by facilitating reconsolidation, the authors administered DCS PRIOR to fear extinction training. Please explain and critically discuss why this timepoint for DCS administration was chosen.

Response: We recognize that a design that mirrors that of the clinical trial - i.e., pre-dosing vs. post-dosing vs. targeted dosing vs. placebo - would have been most optimal for addressing the question of whether DCS can facilitate fear extinction retention (and under which conditions). At the time of study onset, there was no clear evidence suggesting that post-dosing outperforms pre-dosing. Hence, because this experiment was considered secondary to the trial and we aimed to make an adequately powered experiment feasible, we opted for a two-cell design comparing pre-dosing to placebo. 

4. Figure legends 2 and 3 are sloppy, they just give a title and do not provide information on the data presented and the abbreviations in the figures (kind of data shown (mean +/- sem?). Needs to be revised.

The y-axis of the figure states the measure (e.g., SCR mean +/- SE). We have explained all abbreviations in the figure legend: 

“PBO = placebo group, DCS = d-cycloserine, SCR = skin conductance response, SE = 1 standard error, CS- = stimulus that was not paired with shock, CS+E = stimulus that was paired with shock during conditioning and presented in the extinction phase, CS+U = stimulus that was paired with shock during conditioning but not presented in the extinction phase.”

Minor points:

5. Figure 1. For a better temporal resolution of the experimental protocol, a timeline including the images presented in figure 1 and procedures (pairings details also given below the images) should be included. Please also show habituation period in the timeline.

Response: We have revised the figure to better represent the temporal aspects of the procedure.

6. In general, some parts of the MS are written in present tense rather than past tense. Please revise.

Response: We have revised accordingly.

7. On pp8, line 159 check description of the safe context. Furthermore, on the same page, on line 166 I think the contexts for renewal got mixed up and it should be “within threat context” rather than “safe” context. Please correct.

Response: The descriptions of the safe and threat contexts throughout the different stages of the paradigm are now accurate. 

8. Please clarify the abbreviation PBO mentioned for the first time on pp9, line 194. Please either stick to PBO or placebo throughout the manuscript.

Response: We have clarified the abbreviation and used PBO throughout the manuscript

9. Table 1 and 2 seem to be redundancies of the main text (pp11-15)

Response: Our preference is to retain them because they provide the statistics of all model terms, including nonsignificant results, whereas the main text summarizes only the significant results.

---

## [Decision Letter · Decision Letter 1]

20 Sep 2019

PONE-D-19-15283R1

Effect of d-cycloserine on fear extinction training in adults with social anxiety disorder

PLOS ONE

Dear Dr. Hofmann,

Thank you for submitting your manuscript to PLOS ONE. After careful consideration, we feel that it has merit but does not fully meet PLOS ONE’s publication criteria as it currently stands. Therefore, we invite you to submit a revised version of the manuscript that addresses the points raised during the review process. Please find below one small comment of the reviewer.

We would appreciate receiving your revised manuscript by Nov 04 2019 11:59PM. To enhance the reproducibility of your results, we recommend that if applicable you deposit your laboratory protocols in protocols.io, where a protocol can be assigned its own identifier (DOI) such that it can be cited independently in the future. For instructions see: http://journals.plos.org/plosone/s/submission-guidelines#loc-laboratory-protocols

We look forward to receiving your revised manuscript.

Kind regards,

Judith Homberg

Academic Editor

PLOS ONE

Reviewers' comments:

Reviewer's Responses to Questions

**Comments to the Author**

1. If the authors have adequately addressed your comments raised in a previous round of review and you feel that this manuscript is now acceptable for publication, you may indicate that here to bypass the “Comments to the Author” section, enter your conflict of interest statement in the “Confidential to Editor” section, and submit your "Accept" recommendation.

Reviewer #1: (No Response)

2. Is the manuscript technically sound, and do the data support the conclusions?

Reviewer #1: (No Response)

3. Has the statistical analysis been performed appropriately and rigorously? 

Reviewer #1: (No Response)

4. Have the authors made all data underlying the findings in their manuscript fully available?

Reviewer #1: (No Response)

5. Is the manuscript presented in an intelligible fashion and written in standard English?

Reviewer #1: (No Response)

6. Review Comments to the Author

Reviewer #1: The authors have addressed all my earlier comments. I have only one minor comment. The authors argue that the 62.5 % reinforcement rate "... was used ... because lower reinforcement rates create more uncertainty and lead to slower extinction [31,32], thus allowing more room for potential DCS augmentation effects." . As DCS is not believed to facilitate extinction learning, but rather to enhance extinction consolidation, I am not sure whether the second part of their argumentation is valid.

7. PLOS authors have the option to publish the peer review history of their article (what does this mean?). If published, this will include your full peer review and any attached files.

Reviewer #1: No

---

## [Author Response · Author response to Decision Letter 1]

25 Sep 2019

Reviewer #1: The authors have addressed all my earlier comments. I have only one minor comment. The authors argue that the 62.5 % reinforcement rate "... was used ... because lower reinforcement rates create more uncertainty and lead to slower extinction [31,32], thus allowing more room for potential DCS augmentation effects." . As DCS is not believed to facilitate extinction learning, but rather to enhance extinction consolidation, I am not sure whether the second part of their argumentation is valid.

Response: We changed the sentence

“This reinforcement rate was used to replicate procedures from the previously validated paradigm used for this study [21,22], because lower reinforcement rates create more uncertainty and lead to slower extinction [31,32], thus allowing more room for potential DCS augmentation effects.”

To the following:

This reinforcement rate was used to replicate procedures from the previously validated paradigm used for this study [21,22], and to prevent the rapid extinction seen in protocols with 100% reinforcement [27,31].

---

## [Editor Report · Decision Letter 2]

27 Sep 2019

Effect of d-cycloserine on fear extinction training in adults with social anxiety disorder

PONE-D-19-15283R2

Dear Dr. Hofmann,

We are pleased to inform you that your manuscript has been judged scientifically suitable for publication and will be formally accepted for publication once it complies with all outstanding technical requirements.

With kind regards,

Judith Homberg

Academic Editor

PLOS ONE
---

## [Editor Report · Acceptance letter]

9 Oct 2019

PONE-D-19-15283R2 

Effect of d-cycloserine on fear extinction training in adults with social anxiety disorder 

Dear Dr. Hofmann:

I am pleased to inform you that your manuscript has been deemed suitable for publication in PLOS ONE. Congratulations! Your manuscript is now with our production department. 

With kind regards,

on behalf of

Dr. Judith Homberg 

Academic Editor

PLOS ONE